# Two 3D Fractal-Based Approaches for Topographical Characterization: Richardson Patchwork versus Sdr

**DOI:** 10.3390/ma17102386

**Published:** 2024-05-16

**Authors:** François Berkmans, Julie Lemesle, Robin Guibert, Michał Wieczorowski, Christopher Brown, Maxence Bigerelle

**Affiliations:** 1CNRS, UMR 8201-LAMIH-Laboratoire d’Automatique de Mécanique et d’Informatique Industrielles et Humaines, University Polytechnique Hauts-de-France, F-59313 Valenciennes, France; francois.berkmans@uphf.fr (F.B.); robin.guibert@uphf.fr (R.G.); 2Institute of Mechanical Technology, Poznan University of Technology, Plac Marii Sklodowskiej-Curie 5, 60-965 Poznan, Poland; michal.wieczorowski@put.poznan.pl; 3LARSH—Laboratoire de Recherche Societes & Humanites, INSA Hauts-de-France, University Polytechnique Hauts-de-France, F-59313 Valenciennes, France; 4VALUTEC, University Polytechnique Hauts-de-France, CEDEX 9, F-59314 Valenciennes, France; julie.lemesle@uphf.fr; 5U.R Concept, 59300 Valenciennes, France; 6Surface Metrology Laboratory, Worcester Polytechnic Institute, Worcester, MA 01609, USA; brown@wpi.edu

**Keywords:** multiscale analysis, surface topography, fractal-based analysis, Sdr parameter

## Abstract

Various methods exist for multiscale characterization of surface topographies, each offering unique insights and applications. The study focuses on fractal-based approaches, distinguishing themselves by leveraging fractals to analyze surface complexity. Specifically, the Richardson Patchwork method, used in the ASME B46.1 and ISO 25178 standards, is compared to the Sdr parameter derived from ISO 25178-2, with a low-pass Gaussian filter for multiscale characterization. The comparison is performed from the relative area calculated on topographies of TA6V samples grit blasted with different pressures and blasting materials (media). The surfaces obtained by grit blasting have fractal-like characteristics over the scales studied, enabling the analysis of area development at multiple levels based on pressure and media. The relative area is similar for both methods, regardless of the complexity of the topographies. The relevance scale for each calculation method that significantly represents the effect of grit blasting pressure on the increased value of the relative area is a tiling of 7657.64 µm² of triangle area for the Patchwork method and a 124.6 µm cut-off for the low-pass Gaussian filter of the Sdr method. These results could facilitate a standard, friendly, new fractal method for multiscale characterization of the relative area.

## 1. Introduction

Multiscale characterization enables the analysis of surface features present at different scales and facilitates a functional understanding of the relationships between the processing and performance of a surface and its topography. Considering that geometric properties of rough surfaces can differ considerably, there are several multiscale calculation methods [1]. For each method, the application and insights regarding a surface can vary. The use of different multiscale characterization methods depends on several factors. Firstly, the geometry of surface topography may favor one approach over another, as in the case between isotropic and anisotropic surfaces [2,3]. Furthermore, the variety of research domains reflects specific needs that may influence the choice of multiscale calculation methods, i.e., the nature of the geometric characterization should be pertinent to the application [4]. Lastly, choices may be related to algorithmic complexity, which can significantly lengthen characterization depending on the method used. Guibert et al. [5] provided a comparison between three of these methods, discussing the advantages and disadvantages of each method for characterizing polymer abrasion. Multiscale characterization methods, based on fractals or not, employ different mathematical principles to analyze phenomena at various scales. Wavelets, Fourrier series decompositions, power spectral densities and bandpass filters are some examples of non-fractal methods as summarized in the review written by Brown et al. [1]. In this paper, fractal-based methods [6,7] are specifically studied.

The term ‘fractal’, introduced by Mandelbrot in 1975, is used to describe surface topographies that are continuous but not differentiable, with a self-similar or self-affine structure relative to scale [8]. Fractals have shapes or features that can be iterated at different scales. The surfaces of objects can be described mathematically using classical geometry formulas. However, at finer scales, the surface microgeometries can become stochastic and self-similar, like collections of littler scratches on bigger ones, suggesting characterizations through recipes or recursive algorithms. To model and characterize stochastic surfaces, it is necessary to use fractal models to determine the fractal dimensions of topographies. The fractal dimension is used to characterize the complexity of surface topographies. Different methods have been developed to determine different kinds of fractal dimensions [9,10,11]. Fractal methods have been used to simulate chaotic surfaces and to characterize measured surfaces. These models are used to model surface interactions. One method is based on the Weierstrass–Mandelbrot function [12,13,14,15]. This function provides a simulated surface with adjustable parameters, allowing for the desired complexity in applications modeling the size and number of multiscale contacts [13]. In surface analysis, it is possible to analyze the fractal dimensions of surfaces in the forms of 2D or 3D profiles. The methods for analyzing 2D profiles are called length–scale, Richardson or coastline analyses, and it is possible to calculate the relative length. Richardson’s study on the coastlines of Britain, later expanded upon by Mandelbrot [16], is a well-known example. In summary, relative lengths depend on the scale of the observation or calculation. To calculate the relative length of a 2D surface profile, the scale is determined by the size of compasses or dividers that follow the surface profile (Figure 1). The smaller the size of these dividers, the greater their number becomes, allowing them to calculate more details on the profile. The relative length is the ratio of the measured to the nominal length. Relative lengths can be represented on a log–log plot against scale (Figure 2). As the scale decreases, the relative lengths begin to deviate from unity. This deviation occurs when the line segments become short compared to the topographical features, causing significant tilting when they land on the valleys and peaks of the profile. When the profile exhibits self-similarity across a range of scales, the logarithm of the relative length shows a linear increase as the logarithm of the scale decreases. The length–scale fractal dimension is determined by subtracting the slope of the length–scale plot from the unity, as specified in ASME B46.1.

To understand the difference between the length scale and area scale, we can conduct a similar study to that of the coastlines of Britain, but this time trying to calculate the area of a mountain land. Area scale analysis involves calculating the areas of surfaces at various scales. Following Richardson’s and Mandelbrot’s methods, Brown developed a method for calculating the relative area using a 3D triangular tiling with the same philosophy as the relative length with scale variation. Area scale analysis is a type of fractal multiscale analysis. Surfaces containing chaotic elements exhibit scale-dependent variations in their surface areas. The importance of area in understanding performance is emphasized by the fact that many interactions that impact physical functionality are area-dependent. This observation emphasizes the potential of area scale analysis in distinguishing surfaces with different behaviors and in correlating with performance and behavior. The characterization of surface topographies nowadays leans more towards an areal analysis of the surface [17]. Industrial requirements necessitate a deeper understanding of surface features for effective analysis. This goes beyond the limited use of a straight axis for characterizing 2D profiles. To define the development of the relative area of a complex surface (e.g., after sandblasting), where craters may nest within larger craters in a self-similar manner, the fractal area scale method is well suited.

This study will present two methods for calculating the relative area, in line with the fractal philosophy of area scale computation from Richardson to Mandelbrot: the developed interfacial area ratio (Sdr) parameter (ISO 25178-2 [18]) using a low-pass Gaussian filter (ISO 16610-61 [19]) and the triangular tiling method or Patchwork method. The aim is to introduce a technique for computing the relative area, which leverages two elements commonly found in standards: the Sdr parameter as defined in ISO 25178-2 and a set of low-pass Gaussian filters. The idea is to iterate the calculation of the Sdr parameter using filters to change the scale of calculation. The advantage would be to increase the calculation speed of the relative area for multiscale characterization and to use elements already present in existing surface processing software such as MountainsMap^®^ version 9. This new method would allow for an expansion of the scope of application, as multiscale characterization methods depend on the nature of the surfaces, i.e., a certain method is more suitable for a given surface.

## 2. Materials and Methods

### 2.1. Materials

Considering that these characterization methods (Patchwork and Sdr) serve to quantify the developed area of a surface topography, it was necessary to find a way to control the topography using the same process to avoid introducing bias into our further statistical methods. The manufacturing process of the samples and the increasing developed area needed to be correlated to ensure control over the experiment. For this reason, this study presents grit-blasted TA6V logs. Using two factors, namely the grit-blasting media and the pressure of the blaster, it was possible to create a wide range of surfaces and to influence the areal increase due to surface work hardening.

The dimensions of the TA6V logs were a 30 mm diameter and 20 mm height, and they were ground with SiC papers from grit 80 to 4000 before grit blasting. An indentation test was performed on this material to determine its mechanical properties. Ground TA6V surfaces were therefore grit blasted using the Guyson Euroblast 6SF system. Three grit materials were used to blast the TA6V logs:two types of micro balls of glass silico–soda–calcium (G 100 (particle size of 70–150 µm) and G 250 (150–250 µm)) from ARENA;one abrasion material, named C 300 50/80 (particle size of 100–630 µm) from Semanaz, which was composed of hard, sharp, abrasive crystals manufactured from molten glass mass whose material composition was silicate, alumina and iron oxide.

For each grit material, seven pressures were applied from 2 to 8 bar. A total of 35 TA6V logs were blasted, one set for C 300, one set for G 100 and three sets for G 250, to study the repeatability of the grit blasting process. During grit blasting, the blasting gun/log distance was around 10 cm. The grit materials were shot perpendicularly to the TA6V surface during around 30 s for the pressures from 3 to 8 bar and around one minute for the 2 bar pressure to homogeneously blast the whole surface. The grit materials were shot according to a back-and-forth movement (left to right) from the top to bottom of the surface. The 7 grit blasting pressures allowed for a wide variation in relative surface area. The question of the relevant scale for analyzing this process helped in determining which calculation method presented in this study best discriminated the pressure during the grit blasting.

### 2.2. Topographical Measurement and Data Post-Processing

Each blasted TA6V surface was measured by white light interferometry with Bruker ContourGT™ (San Jose, CA, USA). A 50× lens was used which corresponded to an elementary image of 127 × 94.9 µm, and 50 zones of 1 × 1 mm^2^ (5059 × 5058 pixels, 0.198 µm X/Y resolution) were measured randomly on each surface using stitching (540 elementary images, i.e., 27 rows × 20 columns). A total of 1750 measurements were obtained for the 35 surfaces. The surfaces were post-processed and filtered with the software MountainsMap^®^ (Digital Surf^TM^, Besançon, France). Figure 3 shows the measurements of surface topographies. We exhibit a sample of 3 different pressures out of the 7.

### 2.3. Fractal Multiscale Characterization Methods

#### 2.3.1. Method n°1: Patchwork

The first method based on the principle of a developed area is the triangular tiling method, also known as the Patchwork method, which was developed by Brown in the early 1990s [20]. The area, as a function of scale, is determined through a virtual tiling algorithm, such as the one employed in the length–scale analysis (i.e., the coast of Britain). Unlike the length–scale analysis however, which focuses on tiles with line segments, area scale analysis utilizes triangles. Each triangle area serves as a representation of the scale of the calculation. In each tiling instance, all virtual triangles used for tiling have the same area in three dimensions. This places this technique among the methods of fractal analysis based on areal scale. However, when projected onto a datum or nominal XY-plane, the area of these triangles will vary depending on their inclination. The tiling algorithm used in these examples aligns the vertices on the tiling triangles with one of two active rows or columns of heights. These active rows or columns are separated in the *X* or *Y* directions by a distance which is the square root of two times the area of the tiling triangle (Figure 4).

The Patchwork method uses linear interpolation to precisely position vertices along rows or columns, enabling the creation of triangles with desired areas in a 3D tiling process. It begins by setting initial heights for the first triangle and then interpolates the remaining vertices on a similar scale for subsequent triangles within rows or columns. This tiling process can commence from any corner and progress along rows or columns. The outcomes can then be averaged. At larger scales, this method results in the use of more of the measured heights, thereby potentially offering a more accurate representation of the area at that scale. In 2002, the Patchwork method was introduced into the US standards for defining surface textures, ASME B46.1 [21]. Subsequently, in 2012, area scale analysis was incorporated into ISO 25178-2. However, Brown recommends prioritizing using the method presented in ASME B46 [6]. These methods have been applied in several cases, but it is possible to summarize this by two studies: the complexity of the surface (i.e., fractal dimension) of food impacts how the frying process will occur, as shown by Moreno et al. [22], and the fractality of chocolate using the Patchwork method [19].

#### 2.3.2. Method n°2: Sdr Parameter

##### Developed Area Principle

The two calculation methods presented in this study are two approaches that initially allow for quantifying roughness through its correlation between the topography of the surface measured with a microscopy system and the projected surface. This ratio is called the developed area (Rs). The principle is summarized in the study of Lange et al. [23], and it is a foundation for quantifying roughness. To calculate a surface area, one must first compute the sum of the areas of elements defined by four adjacent pixels over an entire measurement. To find the parameter Rs, one must divide the sum of the areas of the elements by the projected area. The parameter Rs can be calculated using Equation (1).
(1)$$Rs=\frac{actual\;surface\;area}{projected\;surface\;area}$$

Plane geometry is employed to determine the surface area of each element. A representative element of area is depicted in Figure 5. The height levels (z) of four adjacent pixels are labeled Z_1_–Z_4_. The line segments between points are designated as S_12_, S_23_, S_34_, S_41_ and S_13_. These line segments constitute the sides of two triangles, the areas of which can be calculated. The sum of the two triangular areas offers an approximation of the actual surface bounded by the four adjacent pixels.

##### Sdr Calculation

The Sdr parameter used in this study is a hybrid parameter from the standard ISO 25178-2 [18]. Hybrid parameters use both information present in elevations and their positions to a similar extent. Examples of such hybrid parameters include the arithmetic mean slope, the root mean square slope, the arithmetic mean summit curvature and the area ratio. Hybrid parameters are highly sensitive to scale and their values are influenced by the data resolution [24]. The Sdr parameter calculates the ratio of the incrementation of the developed surface to the sampled surface. The ratio of the developed interfacial area reflects the combined characteristics of surfaces. A high value of this parameter indicates the importance of either the amplitude, spacing or both [25]. The analysis of the Sdr parameter is relevant for studies on wettability, coating and conductivity in the electronics industry. For wettability, according to the study of Werb et al. [26], since the relative increase in total surface area is closely linked to wetting energy, it is expected that this parameter can effectively differentiate between biofilm variants. Initially, the Sdr parameter (ISO 25178-2) is calculated according to Equation (2).
(2)$$Sdr=\frac{1}{A}\left[\iint_{A}\left(\sqrt{\left[1+{\left(\frac{\partial z\left(x,y\right)}{\partial x}\right)}^{2}+{\left(\frac{\partial z\left(x,y\right)}{\partial y}\right)}^{2}\right]}-1\right)dxdy\right]$$

The Sdr parameter can be expressed as a dimensionless positive number or as a percentage. For instance, a flat and smooth surface would have a value of zero. Essentially, the parameter serves as an indicator of a surface’s complexity and is particularly valuable for tracking surface changes across different processing stages. This characteristic also makes it beneficial for adhesion applications. It is important to mention that the parameter is significantly affected by the sampling scheme, including the number of points and the spacing in the *X* and *Y* axes [27]. However, a comment needs to be made as the following: the formula presented in the standard implies that the surface is differentiable everywhere, which is not the case with fractal surfaces, as they may exhibit singularities and abrupt variations that are not represented by differentiable functions.

The calculation of Sdr shows a similarity with Equation (1) for calculating the Rs but the difference is that Sdr uses the mean value of two triangulations (Figure 6) and not only one as for the Rs [25] (Equation (3)). The provided equation calculates a representative value, denoted as *A_i,j_*, for a specific cell in a grid or matrix. In this formula, the distances between the points *A, B, C* and *D* in a quadrilateral are computed using vectors. Subsequently, the average lengths of the quadrilateral sides are calculated to obtain the final value. This average is detailed in an expression utilizing the coordinates (*x*, *y*) of each point to compute the Euclidean distance between them.
(3)$$\begin{array}{l} A_{i,j}=\frac{1}{2}\;\;\left[\left(\;\frac{1}{2}\left|\vec{AB}\cdot \vec{AD}\right|+\frac{1}{2}\left|\vec{CB}\cdot \vec{CD}\right|\right)+\left(\frac{1}{2}\;\left|\vec{BA}\cdot \vec{BC}\right|+\frac{1}{2}\left|\vec{DA}\cdot \vec{DC}\right|\right)\right]\\ =\frac{1}{4}\left(\left|\vec{AB}\right|+\left|\vec{CD}\right|+\left|\vec{AD}\right|+\left|\vec{BC}\right|\right)\\ =\frac{1}{4}\;\left\{\begin{array}{c} \left({\left[∆y^{2}+{\left(z\left(x_{i},y_{j}\right)-z\left(x_{i,}y_{j+1}\right)\right)}^{2}\right]}^{\frac{1}{2}}+{\left[∆y^{2}+{\left(z\left(x_{i+1},y_{j+1}\right)-z\left(x_{i+1,}y_{j}\right)\right)}^{2}\right]}^{\frac{1}{2}}\right).\;\\ \left({\left[∆x^{2}+{\left(z\left(x_{i},y_{j}\right)-z\left(x_{i+1,}y_{j}\right)\right)}^{2}\right]}^{\frac{1}{2}}+{\left[∆y^{2}+{\left(z\left(x_{i},y_{j+1}\right)-z\left(x_{i+1,}y_{j+1}\right)\right)}^{2}\right]}^{\frac{1}{2}}\right) \end{array}\right\} \end{array}$$

The calculation of the developed interfacial area ratio, Sdr, is derived directly from the digitized measured dataset, exclusively at the scale of the sampling interval. It is not initially suitable for estimating a fractal dimension. In essence, it does not represent the genuine developed area since this concept is meaningful only when it is associated with the scales of measurement and computation. Given that the calculation of the Sdr parameter does not currently allow it to be used as a multiscale calculation method, modifications were made. The particularity of this study consisted of varying the scale of the topography during the calculation of Sdr by using a low-pass Gaussian filter, in compliance with the ISO 16610-61 [19] standard. Appendix A presents some filtered surfaces used to visualize the topographical changes according to the cut-off length of the low-pass filtering. The difference between the developed area and relative area is semantic. The developed area expresses the ratio of the Sdr parameter taken at a single scale, whereas as we vary the calculation scales of this ratio, this area becomes relative.

#### 2.3.3. Differences between Both Methods

For a more comprehensive understanding of this study, it is important to highlight the major differences between these two methods. This could be achieved by simplifying and schematizing the comparison, not calculating an area on a 3D profile but rather a length on a 2D profile. The calculation of the developed length for Sdr (Figure 7) was performed by following the sampling rate and summing the length between each point. This method is therefore at a constant pace (k∆x). On the other hand, the Patchwork method operates at a constant length; it is possible to modify the number of steps by reducing their size, thereby better fitting the measured profile.

### 2.4. Statistical Analysis

Given the complexity of our factors (pressure, media, scale), directly comparing the means of distributions whose nature was unknown would have been both lengthy and risky. To compare the two methods of the relative area calculation, a robust statistic is required such as the mean and the standard deviation. A bootstrap sampling protocol was therefore used to quantify the variation in distributions of both methods. Bootstrapping is a resampling method that involves drawing repeated samples with replacements from a given dataset to estimate its distribution and assess data variability [28,29]. We replicated the value of the relative area 1000 times for the 50 measurements on all of the TA6V logs. Employing bootstrapping in statistical analysis can offer significant advantages, particularly when one aims to circumvent assumptions about the underlying data distribution, especially in cases where this distribution deviates from a normal distribution. Only based on these assumptions can the correspondence between the Sdr and Patchwork prove to be reliable.

## 3. Results

The relative area values were calculated from the two methods. The first part of the results aims to establish the reliability of our data, and the second part aims to define a relevance scale to measure the impact of sandblasting on surface geometry. Figure 8 presents the relative areas calculated by both methods on the surfaces obtained with the highest pressure (8 bar) and the hardest material (C 300). It can be observed that at large scales, the relative areas are unified regardless of the method. Whether it was tiling or Gaussian filtering, it did not compute the details of the topography at large scales and resulted in minimal or null changes.

To analyze the data distribution, a bootstrapping replication was performed, consisting of 50 measurements on a sample grit blasted with C 300 at 8 bar. Since the results of the relative area varied depending on the size of the triangular area of the tiling, this procedure was repeated three times on different sizes as shown in Figure 9.

The histograms of Figure 9 do not show bias considering the distribution of values for the relative area. It was then possible to analyze the measurement data behavior across all tile sizes. The plot presented in Figure 10a shows the relative area values across all scales of the 50 measurements on the surface obtained by grit blasted with C 300 at 8 bar. The points were replaced by a line to analyze the distribution of each measurement. It can be observed, firstly, that the distribution follows the same trend for each measurement. In Figure 10b, a bootstrap replication was conducted, this time on the relative area measurement values of the 50 measurements, each at every tile scale, echoing the histograms presented in Figure 9. It can be noted that the distribution follows the same trend as Figure 10a, indicative of the stability of the Patchwork method at all scales (i.e., no fundamental changes in the distribution). Figure 10c depicts the average of the curves presented in Figure 10a, compared with Figure 10d, which represents the average of the bootstraps. Both curves are similar, which may indicate that the bootstrapping replication did not significantly alter the mean of the original data. Therefore, it can be assumed that the distribution can give a robust mean.

Finally, a comparison was performed by studying the medians of the distribution values across the 50 measurements by categorizing the calculation method and the pressure (Figure 11). By selecting the medians for each category, it was possible to compare the central tendency among them without being affected by extreme values or differences in dispersion. The first observation drawn from analyzing Figure 11 is the difference in trend regarding material change. There is a greater dispersion of distributions noted for G 100 (Figure 11a) compared to other materials. This is due to the size of the abrasion material G 100 (70–150 µm) which will have a minimal impact on surface topography modification, i.e., the relative area, at low pressures. Conversely, more aggressive materials more easily reach the hardness limit of TA6V due to work hardening, explaining the closer distributions for G 250 (Figure 11b) and even more so for C 300 (Figure 11c). The second observation derived from these graphs (Figure 11) is the systematic correlation between the values calculated using the two methods. The smallest value of the Gaussian filtering cut-off length is about 0.8 µm, which is why the curves of the Sdr method always start from this value. However, the values are still correlated with those of the Patchwork method. The reason why the relative area value at 8 bar (Figure 11b,c) pressure is lower at smaller scales with less pressure is that the work-hardening rate will flatten the surface up to a certain limit, making it a smaller relative area value at smaller scales.

The analysis of the relationship between the blasting pressure and the value of the relative area is consistent. Figure 12 indicates that the distribution of the relative area values varies according to the applied pressure. The higher the pressure, the more complex the surface becomes, and its relative area increases. The distributions between the two calculation methods can be compared at the same scale: the distributions follow the same trends for both methods except for the G 100 media at 7 and 8 bar, but both methods invent for G 250 at 7 and 8 bar. Upon examining the mean and standard deviation values for media G 100 (Table 1a), the means vary slightly across the calculation methods up to bar 7. However, for measurements taken on the sandblasted sample at a pressure of 7 bar, the values differ significantly and are closer at bar 8. The means calculated across all media are slightly lower for the Patchwork method than for the Sdr method, although some exceptions confirm that this is not systematic. The Patchwork method generally exhibits a greater dispersion of data around the mean, which is reflected in slightly higher standard deviations.

## 4. Discussion

The results demonstrate a strong correlation between the two methods of calculating the relative area. However, to express the limitations of this study, we can focus on two aspects. The Patchwork method introduces measurement uncertainties at very small scales. This can potentially be explained when the triangle size for tiling falls below the sampling interval. During the experimentation, no S-Filter was applied to remove microroughness associated with measurement noise, as recommended by ISO 25178-3 [30]. Following the calculation philosophy of the Patchwork method, we believe that triangular tiling at very small scales amplifies measurement noise because it interpolates between measured points. This phenomenon can be observed in Figure 8, where the third tiling exercise indicates a higher relative area value than the previous one, which logically should not be the case. One of our assumptions is that the Patchwork method acts as a low-pass Gaussian filter by removing details at each surface tiling scale. This initial point leads to the second, which concerns measurement uncertainties in general. The fluctuations in heights can be estimated through multiple topographical measurements at the same location, as demonstrated by Lemesle et al. [31,32]. The authors argue that the largest measurement fluctuations correspond to regions with significant plastic deformations, namely from grit blasting. In Appendix B of Lemesle’s study, two graphs depict the fluctuations of the Sdr parameter on surfaces blasted at 3 and 6 bar, measured 100 times. These results indicate that the Sdr parameter fluctuates over time when the surface exhibits a certain level of complexity, suggesting a significant variation in fractal surfaces. Another trend is that the graphs generally show an increase in the relative area as a function of pressure, as indicated in Figure 11. However, in the distributions at 7 and 8 bar in Figure 12b–d, an overlap or even exceeding of the histogram is observed for 7 bar compared to 8 bar. The hypothesis for this is that the grit blaster struggled to maintain a pressure of 8 bar, which was its maximum capacity, and it is possible that the TA6V logs were grit blasted at a lower pressure. If we consider this hypothesis, we can still observe that the rest of the grit blasting process is consistent in the relationship between the relative area and the pressure.

## 5. Conclusions

From the results of the comparison between the Patchwork method and the Sdr method using a low-pass Gaussian filtering, it is possible to observe a strong similarity between the two methods for calculating the relative area. The Sdr method offers several advantages: its computation time is shorter and its components are derived from ISO standards (i.e., computation and filtering). The relevance scale for each calculation method that significatively represents the effect of grit blasting pressure on the increased value of the relative area is a tiling of 7657.64 µm^2^ of triangle area for the Patchwork method and a 124.6 µm cut-off for the low-pass Gaussian filter of the Sdr method. Moreover, these components are already implemented in surface processing software solutions, making them easily accessible to researchers. These findings are important as they demonstrate the robustness and reliability of both approaches for calculating the relative area, providing researchers and practitioners with flexibility in choosing the appropriate method based on the specific needs of their study or application. However, it is important to note that despite the similarity in results, each method has its own advantages and limitations, which should be considered when using them. Therefore, it is recommended for researchers and practitioners to carefully consider the specific characteristics of their samples and the objectives of their study before choosing the appropriate measurement method.

## Figures and Tables

**Figure 1 materials-17-02386-f001:**
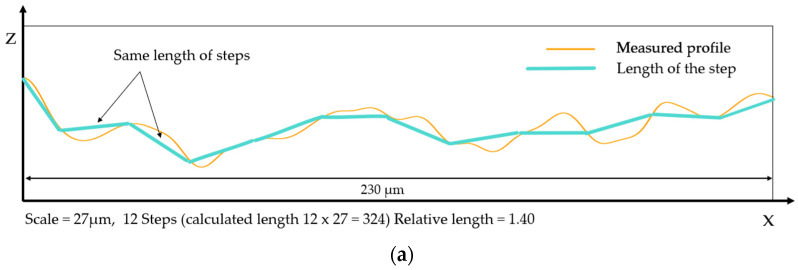
Two calculations of the relative length for two different scales. (**a**) Calculation from a profile view with 4 steps. (**b**) Calculation from a profile view with 12 steps. The calculated length is the sum of the step length multiplied by the number of steps. The nominal length is 230 µm.

**Figure 2 materials-17-02386-f002:**
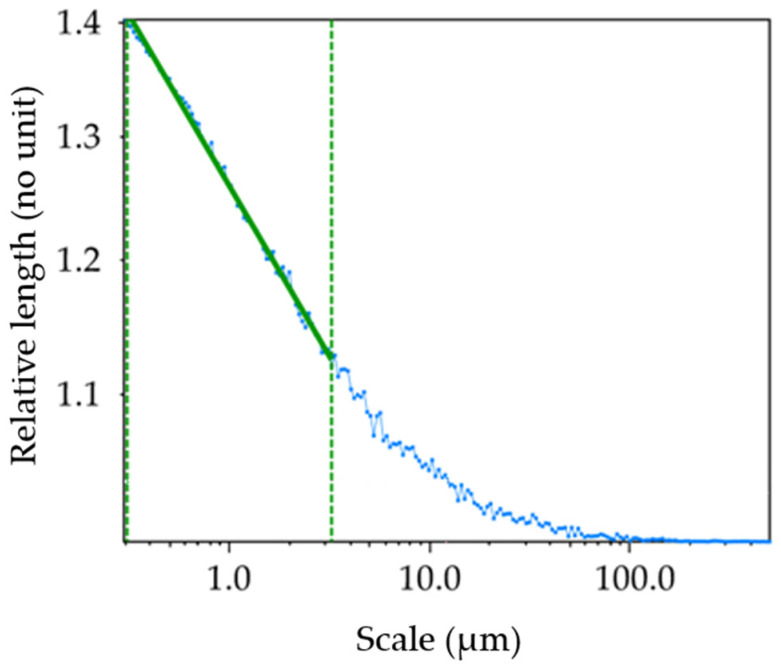
Length–scale plot. The self-similarity over some range of scales is emphasized by the regression line in green. According to ASME B.46, the fractal dimension based on the length scale is 1.095 (no unit). The blue line is the interpolation between the values for the calculation of the relative length on every scale. The dashed green corresponds to the fractal domain, which is the range where the surface is self-similar on different scales.

**Figure 3 materials-17-02386-f003:**
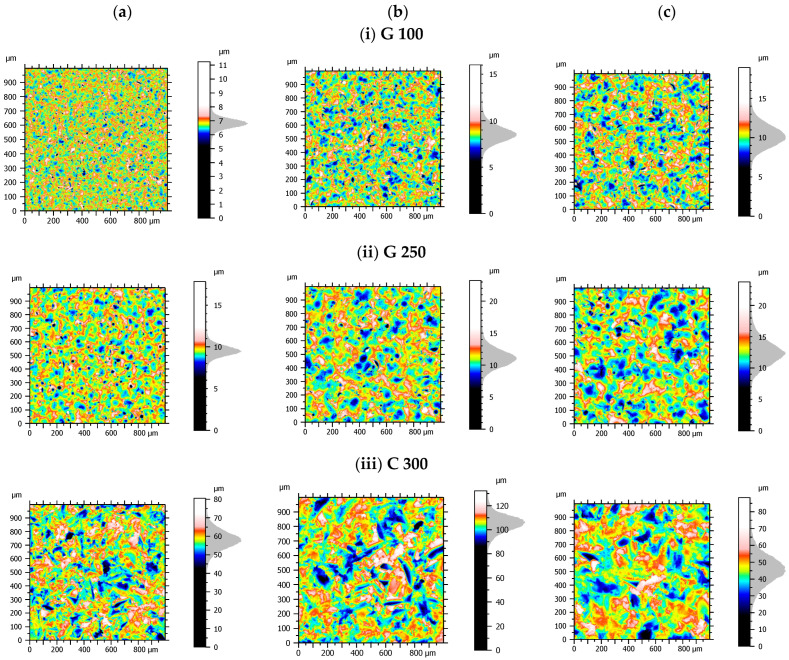
Selection of surface topographies classified by pressure of grit blasting (**a**–**c**) and blasting material (**i**–**iii**). The topographies for the media G 100 (**i**) and G 250 (**ii**) have more circular features considering the spherical nature of the glass beads. The topographies of the C 300 medium (**iii**) have more sharply edged indents due to the angular nature of the corundum.

**Figure 4 materials-17-02386-f004:**
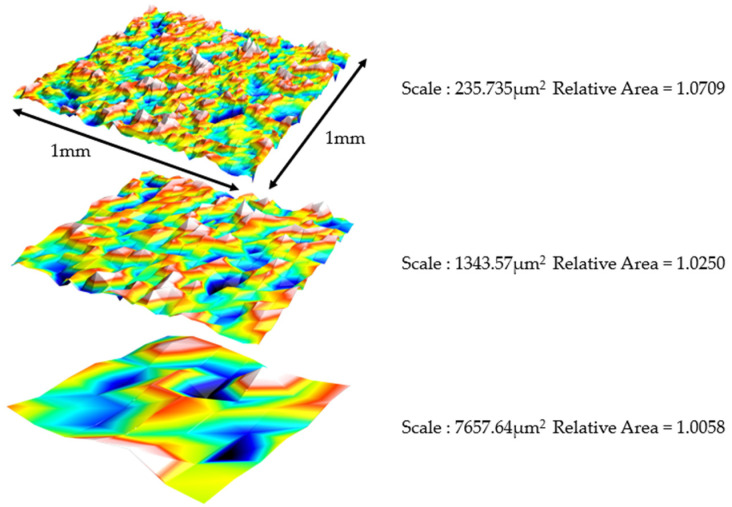
Triangular tiling at three different scales of a TA6V surface grit blasted with the C300 medium and a pressure 8 bar. The scale is the area of the triangular tiles, which have the same area but different projected areas, depending on the inclination.

**Figure 5 materials-17-02386-f005:**
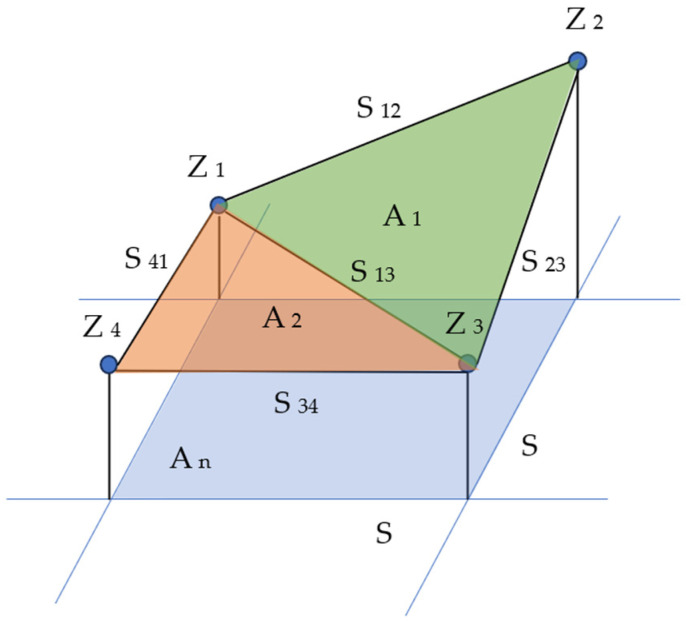
Representation of the four neighboring pixels (Z_1_ to Z_4_) of the surface topography used to create two triangular areas (A_1_ in green and A_2_ in orange) with segments (S_12_, S_23_, S_34_, S_13_, S_41_) and a comparison on the projected area (A_n_ in blue).

**Figure 6 materials-17-02386-f006:**
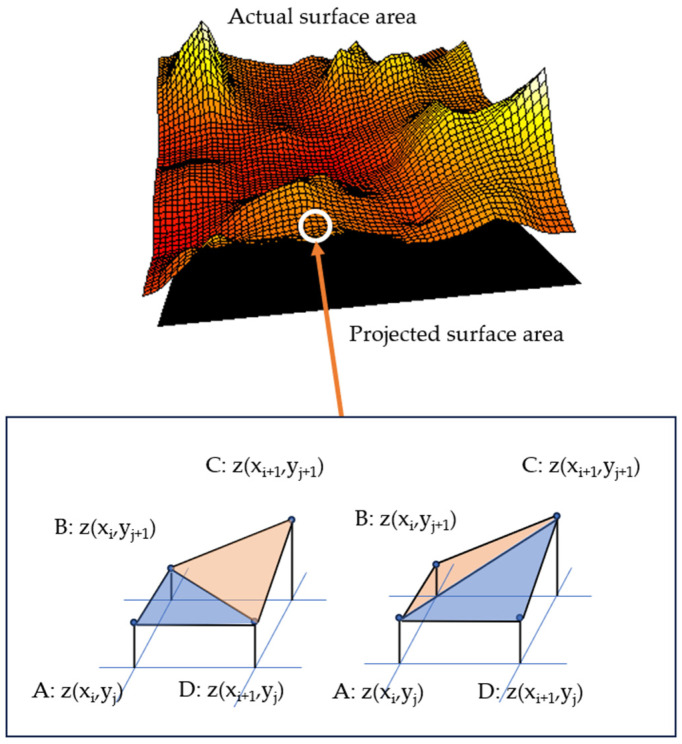
Actual surface area as topography on a given scale (color) on the projected surface area (black). The topography is represented as squares of 4 pixels. The magnification is representing the calculation of the area between four adjacent points (A–D) calculated from the mean value of two triangulations (blue triangles) [25].

**Figure 7 materials-17-02386-f007:**
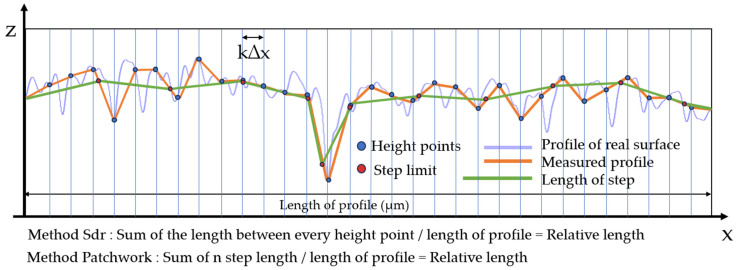
Diagrams of the two calculation methods used in this study for the developed length. The blue line is a representation of a real surface. The orange line is a linear interpolation between measured height points which is our measured profile (the Sdr method was used for computing the relative length at the sampling scale). The green line is a representation of the Patchwork method following the measured profile using the same length steps and sometimes interpolating between measured height points.

**Figure 8 materials-17-02386-f008:**
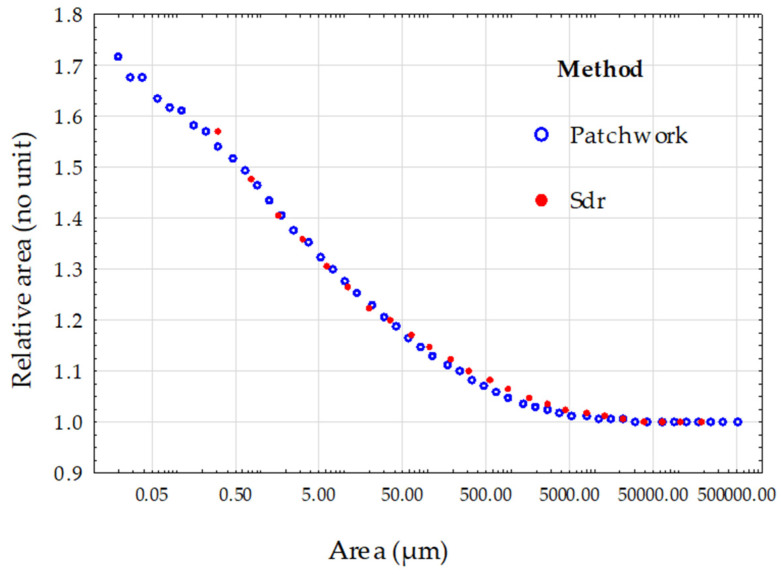
Comparison of the two methods, Sdr and Patchwork, for the calculation of the relative area on the surface topographies created with the C 300 grit-blasting material and a pressure of 8 bar (these are the most aggressive conditions of our material/pressure experimentation). The blue rings represent the values of the relative area calculated by the Patchwork method depending on the size of the triangle tiling (patch area). The red dots represent the calculation of the relative area related to one of the 24 cut-off lengths of the low-pass filter.

**Figure 9 materials-17-02386-f009:**
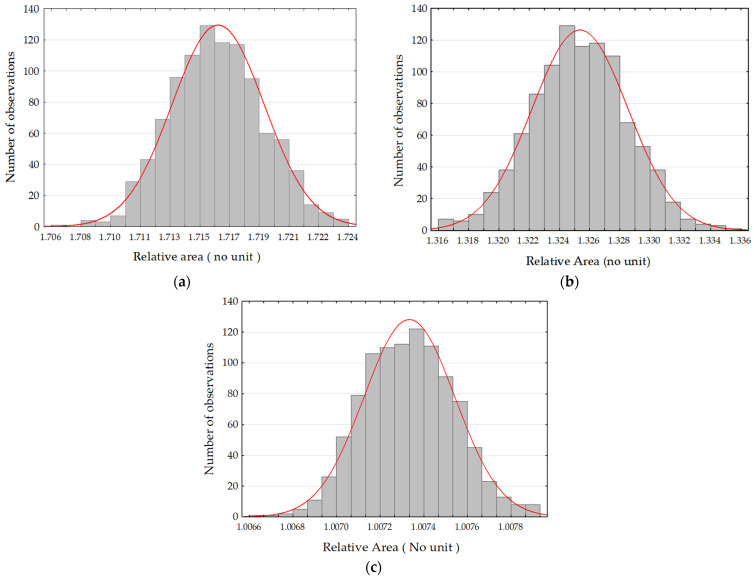
Density probability distributions obtained after applying the bootstrapping protocol to the relative area measurement data. The area of the tiling triangles corresponds to (**a**) 0.02 µm^2^, (**b**) 5.124 µm^2^ and (**c**) 10,845 µm^2^.

**Figure 10 materials-17-02386-f010:**
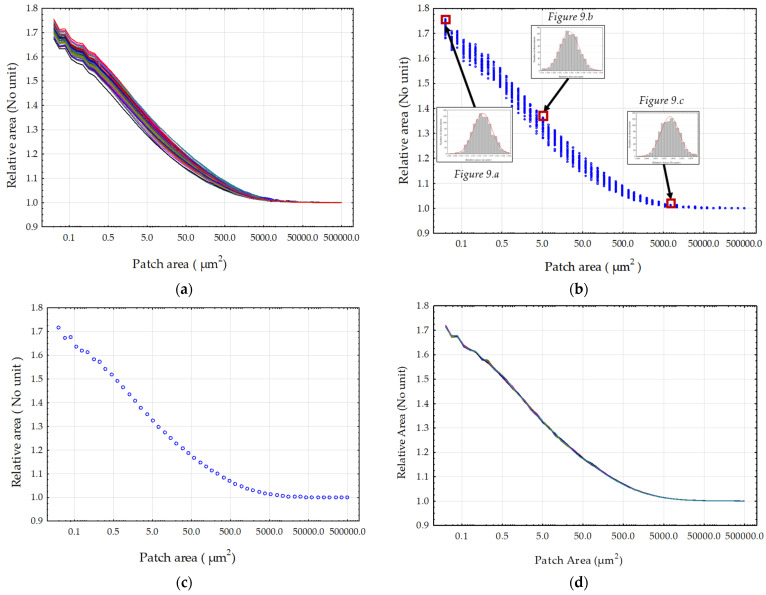
Distributions of the relative area values calculated by the Patchwork method from the 50 measurements of the sample sandblasted at 8 bar with C 300: (**a**) the lines of the 50 sample measurements, (**b**) the values after resampling by bootstrapping, (**c**) the averages of the original measurements and (**d**) the averages of the bootstrapped values.

**Figure 11 materials-17-02386-f011:**
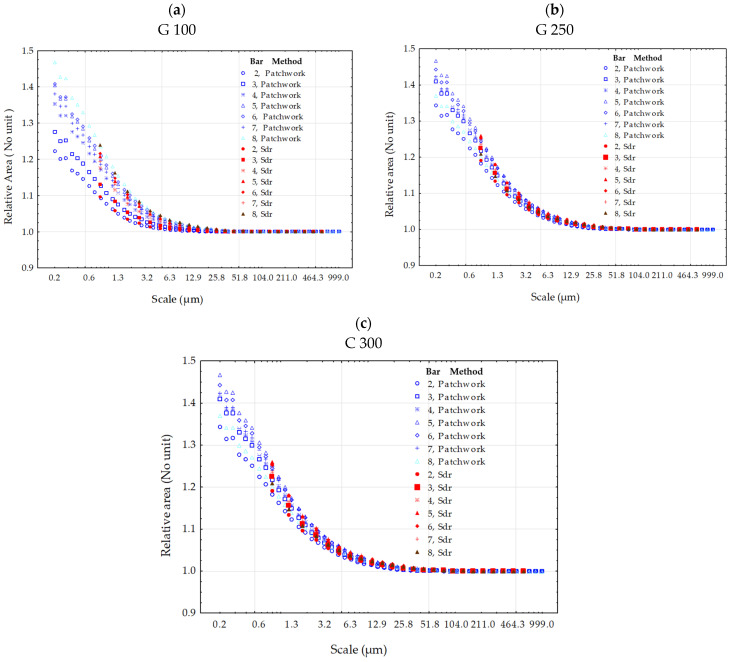
Comparison of the two methods, Sdr and Patchwork, for the calculation of the relative area of the surface topographies created with the grit-blasting materials G 100 (**a**), G 250 (**b**) and C 300 (**c**). The points represent the medians of the distribution of the relative area values, categorized by the calculation method and pressure. The blue symbols represent the median points for the Patchwork method and the red symbols correspond to the Sdr method. The scale references the cut-off length for the low-pass Gaussian filter applied for the Sdr calculation. In this scale, the tiling size in µm^2^ for the Patchwork method is equal to the square of the cut-off length divided by 2.

**Figure 12 materials-17-02386-f012:**
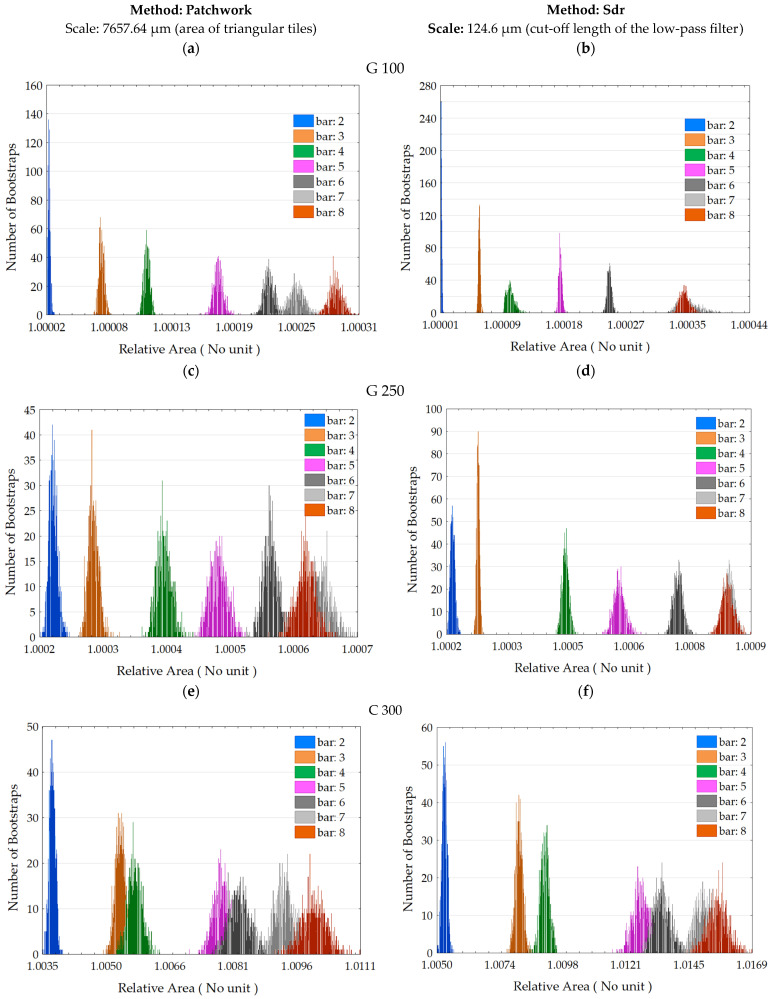
Distributions of the relative area values by method of calculation (Patchwork (**a**,**c**,**e**) and Sdr (**b**,**d**,**f**)), media (G 100 (**a**,**b**), G 250 (**c**,**d**) and C 300 (**e**,**f**)) and pressure.

**Table 1 materials-17-02386-t001:** Means and standard deviations of the relative area from distributions of Figure 12, by media (G 100, G 250, C 300), pressure (2 to 8 bar) and method of calculation (Patchwork, Sdr).

(a) G 100				
Pressure (bar)	Relative area			
	Method: Patchwork		Method: Sdr	
	Mean	Standard deviation	Mean	Standard deviation
2	1.000023	1.17 × 10^−6^	1.000008	6.67 × 10^−7^
3	1.000071	2.51 × 10^−6^	1.000061	1.40 × 10^−6^
4	1.000112	2.83 × 10^−6^	1.000104	5.52 × 10^−6^
5	1.000178	4.49 × 10^−6^	1.000174	2.30 × 10^−6^
6	1.000224	4.61 × 10^−6^	1.000242	3.10 × 10^−6^
7	1.000250	7.44 × 10^−6^	1.000352	1.49 × 10^−6^
8	1.000286	6.15 × 10^−6^	1.000346	6.64 × 10^−6^
(b) G 250				
Pressure (bar)	Relative area			
	Method: Patchwork		Method: Sdr	
	Mean	Standard deviation	Mean	Standard deviation
2	1.00018	6.97 × 10^−6^	1.00017	5.79 × 10^−6^
3	1.00024	8.43 × 10^−6^	1.00024	3.64 × 10^−6^
4	1.00036	1.16× 10^−5^	1.00045	8.36 × 10^−6^
5	1.00045	1.42 × 10^−5^	1.00059	1.45 × 10^−5^
6	1.00053	1.21 × 10^−5^	1.00074	1.16 × 10^−5^
7	1.00060	1.97 × 10^−5^	1.00087	1.20 × 10^−5^
8	1.00058	1.52 × 10^−5^	1.00086	1.52 × 10^−5^
(c) C 300				
Pressure (bar)	Relative area			
	Method: Patchwork		Method: Sdr	
	Mean	Standard deviation	Mean	Standard deviation
2	1.0038	7.22 × 10^−5^	1.0053	9.72 × 10^−5^
3	1.0054	1.16 × 10^−4^	1.0081	1.41 × 10^−4^
4	1.0057	1.69 × 10^−4^	1.0091	1.68 × 10^−4^
5	1.0078	2.11 × 10^−4^	1.0127	3.19 × 10^−4^
6	1.0082	2.40 × 10^−4^	1.0135	3.17 × 10^−4^
7	1.0093	2.35 × 10^−4^	1.0151	3.65 × 10^−4^
8	1.0100	2.94 × 10^−4^	1.0156	3.64 × 10^−4^

## Data Availability

Data are contained within the article.

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
