# Peer review of "Two 3D Fractal-Based Approaches for Topographical Characterization: Richardson Patchwork versus Sdr"

_materials, 2024, doi:10.3390/ma17102386_

Round 1

Reviewer 1 Report

Comments and Suggestions for Authors

The manuscript entitled “Two 3D fractal-based approaches for topographical characterization: Richardson patchwork versus Sdr” is well-written. It is very informative on the topic of surface topographical characterization with rich details on the background knowledge of the topic. I believe this manuscript will be an excellent reference for readers who are interested in surface topographical characterization.

I have a few comments for the authors for revising the manuscript, as below.

1. There is a mistake in Figure 1. The red line in the figure legend with the label “Measured profil” should be orange, not red. In addition, I think “profil” should be replaced with “profile”.

2. Figure 2 misses some necessary details: the meanings of the dashed green, dashed red and solid blue lines.

3. In Figure 2, the units for the x- and y-axes should be indicated on the x- and y-labels respectively.

4. There is a typo in Line 150.

This manuscript does not have many typos, but I still suggest the authors to carefully and thoroughly check the whole manuscript to correct any possible typos.

5. The sentence “The objective is … low pass Gaussian filters.” in Lines 148-150 is difficult to understand and suggested to be rewritten.

6. Line 251: “Figure 3” should be “Figure 5”.

Please note, there are “many” typos throughout the manuscript similar to this one. The figures are not mentioned or are mistakenly mentioned in the manuscript. Please find and revise them all.

7. In Equation 2, the symbol “delta” should be “partial”.

8. The details for explaining Equation 3 should be provided.

Reviewer 2 Report

Comments and Suggestions for Authors

The paper analyzes two methods for the topographical characterization of the complexity of the surface through the 3D approach based on fractals, taking into account the ASME and ISO standards and offering unique insights and applications. Despite all these, there are still some problems to be solved, as follows:

1. For a good understanding of the content of the text of the work, the English language and grammar must be revised.

2. When a notation or symbolization appears for the first time, explain in words what it means and put the symbol/notation in parentheses.

3. There are words that you write once with a dash (e.g. "multi-scale") and other times without a dash (e.g. "multiscale", respectively the method is sometimes written "Richardson patchwork method", sometimes "Richardson Patchwork method", similar to "Relative area" or "Relative Area" (please check throughout the work) etc.

4. In section 1 "Introduction" in the last part you give more information about fractals, and in the first part about the methods of topographical analysis of surfaces. This transition is made suddenly, like a boomerang, without the connection between them emerging!

5. In Fig. 2, in the content the length is 1095 (without measurement unit)!

6. There are some forms that are difficult to understand (for example line 132; "The Area Scale analysis is a form of multiscale analysis"

7. Line 139, how "... contextualizing surface features."!

8. Line 168, "... SiC paper from 80 to 1000 ...", I think it's the grain or what?

9. Line 251 is Fig. 3 or Fig. 5! It is good to check the numbering of the figures in the text, because I don't think they correspond to the actual figures!

10. Fig. 6, nothing is said in the text about it, and section 6 "Results" begins with "Figure 6 ...", but I think it is about Fig. 8! See also in lines 315 and 357!

11. Line 414, as if initially the averages for G100, G250 and C300 were considered at 2, 4 and 8 bar, and now 7 bar also appeared, respectively in Tab. 1 other intermediate values 3, 5, 6, and 7 bar. There is no logical transition between section 2 "Materials and Methods" and Section "Results"! Why were these intermediate pressures necessary (please explain)?

Comments on the Quality of English Language

It needs to be improved!
